# Organocatalyzed bottom-up formation of protocells

Marian Simon Rafael Ebeling [1], Otto Berninghausen [2], Khang Hoang Nguyen [1], Roland Beckmann [2] & Oliver Trapp [1,3] ✉

The organisation of living systems into cellular structures is a characteristic that enables differentiation from the environment. A pivotal step in the development of life is compartmentalisation, achieved through the formation of vesicle-like structures. Fatty acids - or phospholipids - have been used to simulate prebiotic vesicle and protocell formation. However, a process by which amphiphiles are formed from small prebiotically plausible molecules, which spontaneously self-assemble to protocells, is unknown. Here, we demonstrate that an organocatalytic reaction cascade starting from acetaldehyde with prebiotic imidazolidine-4-thione rapidly yields poly(hydroxy) alkenyl aldehydes that spontaneously self-assemble to protocells. In this process, lipid-like molecules (up to C20) develop a membrane, which additionally incorporates the organocatalyst at the liquid-lipid interface. These catalytically active protocells (11 nm – 7 μm) tolerate external influences such as pH value, temperature and salts. This finding unveils an organocatalytic pathway to selective lipid formation and spontaneous compartmentalisation without the necessity of preformed amphiphiles.

Reconstructing the emergence of a self-sustaining and self-reproducing unicellular organism is fundamental to explaining the origin of life under abiotic conditions[1]. Abiogenesis is thought to have begun with a separation from the environment by vesicles because of the advantages of compartmentalisation[2]. Organic molecules can be concentrated in vesicles, leading to a kinetic acceleration of reactions and thus suppressing competing decomposition reactions. Although water is essential as a reaction medium for many processes, water-eliminating reactions require a reaction environment that reduces solvation by confinement in vesicles. This can favour oligomerisation reactions that take place under water elimination, such as the formation of peptides, polysaccharides or information carrying molecules such as DNA and RNA[3,4].

In organisms, fatty acids are synthesised by the fatty acid synthase (FAS)[5], which utilizes a series of decarboxylative CLAISEN condensations from acetyl-CoA and malonyl-CoA (Fig. 1a). To simulate the formation of protocells (Fig. 1b), model vesicle systems with membranes made of monocarboxylic acids[6,7], *N*-acyl amino acids[8,9], alkyl alcohols[10], or phospholipids[11,12] are investigated for their properties to external stimuli such as salts, amino acids, sugars[13] or the inclusion/replication of RNA/DNA[1,11,14–16]. Coacervates[17] are also considered to be independently plausible compartments on early Earth, as aggregation of RNA, DNA or peptides can lead to a membrane-free phase separation[18–22].

Amphiphiles consisting mainly of saturated alkyl chains were proposed to originate from FISCHER-TROPSCH-type (FTT) reactions[23] at high temperature and pressure in geothermal vents[24] or to be of extraterrestrial origin[25], since alkyl carboxylic acids were found in extracts of the Murchinson meteorite[6]. Recently, the highly efficient and robust conversion of $CO_2$ into oxygenated organic compounds (mainly formaldehyde and acetaldehyde) by meteorite and volcanic particles or under photochemical activation[26] has been identified over a wide range of reaction conditions[27]. Unlike the FTT reaction, no long alkanes, which are difficult to activate and convert into long-chain carboxylic acids or alcohols, were found. An alternative pathway towards acetaldehyde was proposed by BONFIO et al., by using UV-C irradiation to decompose α-hydroxythioacetamide into acetaldehyde.

[1]Department of Chemistry, Ludwig-Maximilians-University Munich, Munich, Germany. [2]Department of Biochemistry, Gene Center, Ludwig-Maximilians-University Munich, Munich, Germany. [3]Max Planck Institute for Astronomy, Heidelberg, Germany. ✉e-mail: oliver.trapp@cup.uni-muenchen.de

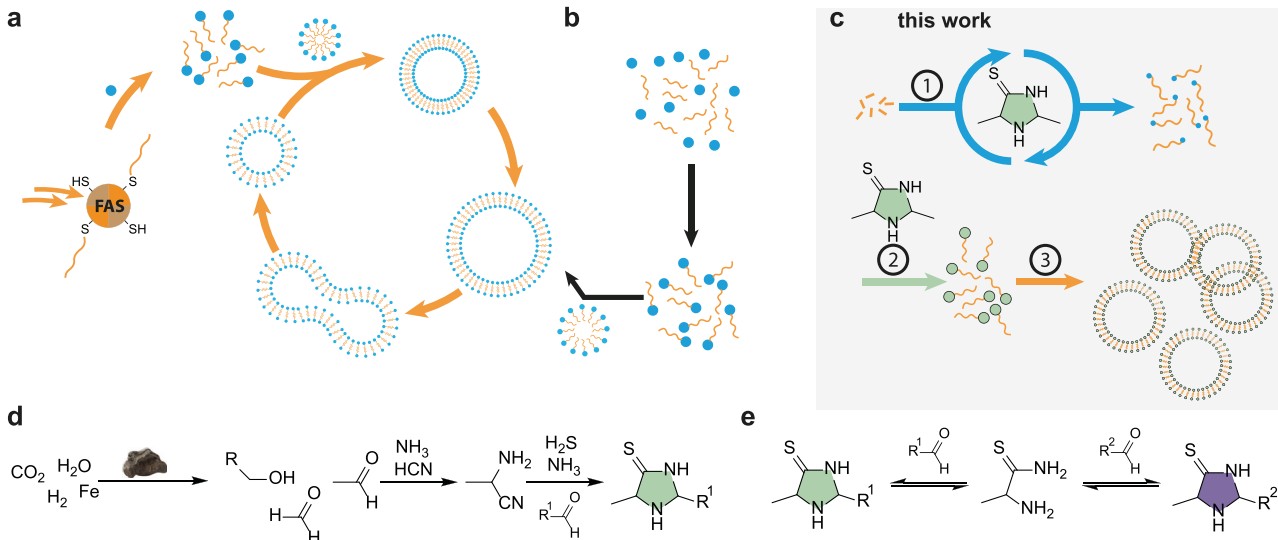

**Fig. 1 | Lipid forming reactions and self-assembly of lipid molecules to vesicles and protocells. a** Biological synthesis of fatty acids by fatty acid synthase (FAS). **b** In-situ generation of artificial protocells using amphiphilic lipids and precursor molecules to release the lipids by reaction. **c** Organocatalytic oligomerisation starting from small molecules (step 1), i.e., acetaldehyde, self-modification of the organocatalyst with the in-situ formed amphiphilic molecules (step 2), and self-assembly to protocells (step 3). **d** Prebiotic synthesis of imidazolidine-4-thione organocatalysts. **e** Dynamic exchange of C-2 aldehyde substituent leading to modified organocatalysts.

In this context they demonstrated a base catalysed aldol condensation of acetaldehyde leading to the formation of unsaturated up to C8-aldehydes[28]. This work also supports the possibility that life may have developed other ways of making membranes that have been adapted to the environment over time[29]. An indicator for this hypothesis are partially unsaturated phospholipids based on geranylgeraniol, which were found in archaea[30]. Most archaean phospholipids consist of partially or fully unsaturated polyprenyl- or terpenoid-type phosphates, which can be considered as remnants of primordial vesicles[31].

In addition to a confined reaction space, catalysis is a key element in the emergence of life. Without enzymes, only catalytically active minerals and metals, e.g., montmorillonite[32], magnetite[33], or reduced iron particles[27,34] were initially available on the early Earth. However, due to their insolubility, reactive heterogeneous surfaces are a poor match for separated reaction spaces. Prebiotic organocatalysts open up a wide range of reactions and can be regarded as precursors of enzymes. They not only allow selective reactions but can also be integrated into vesicles and protocells.

Recently, we found that imidazolidine-4-thiones are selectively formed from H$_2$S, NH$_3$, HCN, and carbonyl compounds under mildly basic aqueous and prebiotically plausible conditions[35]. This class of compounds are thio-MACMILLAN type organocatalysts[36,37] capable of iminium ion or enamine catalysis. These organocatalysts can modify their own building blocks by α-alkylation of aldehydes. Because of the ability to dynamically exchange their substituent at C-2 with carbonyl compounds, they undergo self-modification forming new catalysts with potentially superior catalytic properties. Thus, structural mutation and kinetic selection can occur on the molecular level[38].

In this work, we present a prebiotically plausible organocatalyzed aldol oligomerisation reaction starting from acetaldehyde with imidazolidine-4-thione as catalyst yielding poly(hydroxy)alkenyl aldehydes. These aldehydes and the catalyst form by structural mutation of the catalyst amphiphiles which spontaneously self-assemble to protocells.

## Results and Discussion
While studying prebiotic imidazolidine-4-thione organocatalysts in aldol reactions, we found that, in addition to the activation of carbonyl compounds leading to the expected normal aldol product, aldol-oligomerisation occurs, visible by a turbidity of the reaction solution. In addition, we observed incorporation of the product aldehydes into the imidazolidine-4-thione structure, leading to a modification of the organocatalysts (Fig. 1c).

In particular we found that acetaldehyde, which is formed under prebiotic conditions from CO$_2$ and in situ generated hydrogen by catalysis with meteoritic and volcanic particles, was prone to undergo organocatalyzed oligomerisation[34]. For a comprehensive investigation of this process, we initially selected 2,5-dimethylimidazolidine-4-thione **1** (Fig. 1c) as prebiotic organocatalyst as it is formed in situ in the same mixture from acetaldehyde, NH$_3$, H$_2$S and KCN in water[35] (Fig. 1d). As these substituted imidazolidine-4-thione organocatalysts are chiral and crystallise in enantiopure crystals (conglomerate), we selected (2RS,5S)-2,5-dimethylimidazolidine-4-thione as a prebiotically formed catalyst, which can be derived from L-alanine. In initial screening experiments with varying acetaldehyde and catalyst concentrations, at a slightly acidic pH of 4, we observed that some samples became cloudy over time.

Light scattering (Supplementary Fig. 3) indicated the formation of larger assemblies. Microscopic images revealed the formation of droplets and indicates a correlation between the density of these assemblies and the concentrations of the aldehyde as well as the amounts of catalyst employed. (Fig. 2a, Supplementary Movie 1).

High-resolution Orbitrap mass spectrometry (HRMS) of the suspension revealed the formation of a self-evolving organocatalytic species with extended side chains at the C-2 position or bound as iminium ion/enamine to the secondary amine as the main products, which is an intermediate of the organocatalyzed aldol-oligomerisation (Fig. 1e, Supplementary Figs. 4–5). In addition to chain elongation, unsaturated oligomers were unambiguously identified. These are formed by continuous dehydration of the oligomeric polyols (Supplementary Tables S4–7). As the catalysts can be reversibly hydrolysed by ring-opening, newly formed aldehydes are incorporated by ring closing, resulting in modified organocatalysts with altered physical, chemical and catalytic properties. The product distribution of the oligomers as a function of acetaldehyde and catalyst concentration, identified by HRMS, was systematically mapped by chain length at C-2 against the dehydration steps (Fig. 2b). The initially added organocatalyst **1** with acetaldehyde at C-2 is located on the first data point (chain

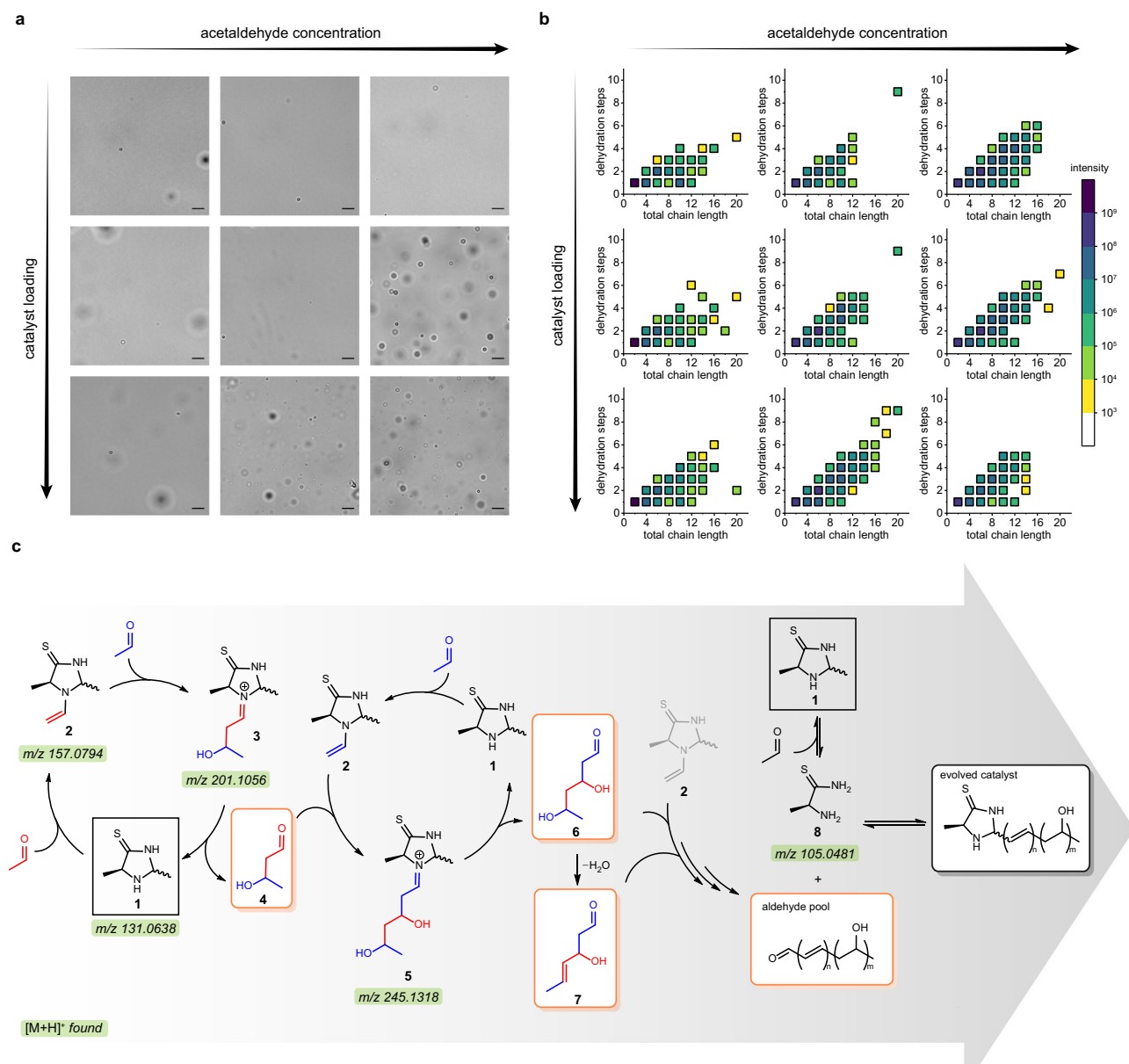

**Fig. 2 | Concentration screening and proposed mechanism of the organocatalysed aldol-oligomerisation and incorporation of the reaction product aldehydes into the organocatalyst. a** Micrographs of an aqueous solution. Scale bar: 10 μm. **b** Extracted HRMS data of catalyst-masses with chain lengths of 2–20 plotted against dehydration grade extracted for $[M + H]^+$. Reaction conditions for a and b: 0.2, 0.5 or 2.0 M acetaldehyde from left to right, with 1, 10 or 20 mol% catalyst loading from top to bottom after 1 d. Sample dilution was referenced to catalyst concentration for every acetaldehyde concentration. **c** Proposed interlocking catalytic cascade of organocatalysed aldol-oligomerisation reaction and molecular self-modification of the catalyst.

length: 2, dehydration steps: 1, Supplementary Fig. 4). This plot gives an overview of product distribution after 1 d. We observed that with increasing chain length, the intensities (amount of product) initially decrease, and the degree of water elimination also increases. The $^1$H NMR data indicate that the formed C = C double bonds are in *E*-configuration ($^3J$ coupling of 15.1 Hz, Supplementary Fig. 9). This was also supported by the selective synthesis of identified intermediates **10a-d** ($^3J$ coupling of 15.3 Hz), which can be accessed by the condensation of aldehyde **7** and **8-HCl** (Fig. 2c, vide infra). An increase in the aldehyde concentration resulted in a shift towards longer chains; compared to an increased catalyst loading, this trend reverses at higher catalyst loading. This leads to the formation of a complex mixture[39] of constitutional and configurational isomers of polyhydroxy aldehydes, poly(hydroxy)alkenyl aldehydes and polyalkenyl aldehydes, visible by overlapping peaks over a broad range in the NMR spectra (3 – 5 ppm,

Supplementary Figs. 10–11). This signal range proves the formation of alkenyl and hydroxy alkenyl compounds and in addition the dynamic exchange of the formed aldehydes with the organocatalyst at the C-2 position, however an unambiguous identification even with the synthesis of reference compounds was not possible. Therefore, we investigated and monitored this chemical process by high-resolution mass spectrometry.

The dynamic process of the oligomerising aldol addition and elimination reaction as well as the exchange reactions with the original organocatalyst **1** is corroborated by the observed intermediates during the reaction. The interlocking catalytic cycles are summarised in Fig. 2c. Acetaldehyde condensates on the secondary amine of the catalyst **1** to form the activated vinyl species **2**. Reaction with a second equivalent of acetaldehyde gives the first aldol addition product **3**. After hydrolysis from the catalyst, the released aldehyde **4** is now part

of the aldehyde pool. It can react in the same way with another vinyl species **2** to elongate the oligomeric chain **5** and be released as aldehyde **6**. Interestingly, the mechanism we propose is like the well-known mitochondrial fatty acid synthesis, in that new $C_2$-units are not inserted at the $CH_3$-end of the chain, but at the beginning, here the aldehyde[40]. Favoured by the lowered pH, an elimination of water to form an unsaturated lipid chain **7** can occur after every aldol addition. As the catalyst is in small equilibrium with the hydrolysed open ring form **8**, free aldehydes can be incorporated. Here, we selectively synthesized aldehyde 7 and cyclised it with **8**-HCl, to form a reference compound **10a-d**. This may result in the natural selection of certain products due to their enhanced stability towards hydrolysis. The newly formed organocatalysts are catalytically active, exhibiting modified selectivity towards the substrate due to modified sidechains, thereby instigating a self-modification process within the system. As already pointed out, we observed a high number of new catalyst species in addition to the original catalyst due to the characteristic NMR signals of the catalyst-ring C-H(3 − 5 ppm, Supplementary Figs. 10−11).

In addition to these findings starting with the preformed 2,5-dimethylimidazolidine-4-thione **1**, we were interested in whether this reaction cascade could start even earlier, in that the organocatalyst itself is first formed and then triggers the aldol-oligomerisation. Therefore, we investigated reactions starting just with thioamide **8** and acetaldehyde under the same experimental reaction conditions. The results show, that the organocatalyst is formed in-situ by ring closing with acetaldehyde, which catalyses the aldol-oligomerisation (Supplementary Figs. 12−13).

A systematic investigation into external factors, i.e. pH dependency and salt concentrations, was performed to gain further mechanistic insight and to characterise the stability of the resulting protocells. The titration curve of the catalyst shows amphoteric characterisics (Supplementary Fig. 2). The determined $pK_a$ values of the thiolactam ($pK_{a1} = 3.8$) and secondary amine ($pK_{a2}$ ~ 11) in aqueous solution showed that the catalyst can buffer the solution at pH 4, which is the optimal reaction condition for the enamine formation. This finding serves to demonstrate the necessity of the thiolactam and underscores its pivotal role in ensuring the maintenance of the optimal pH range. In comparison, lactams are considerably less acidic ($\Delta pK_a = +6$) and therefore cannot buffer the reaction at pH 4[41,42].

When the pH of the reaction was reduced to 2.5, no change in HRMS data was observed, whereas the assembly formation decreased in microscope micrographs. Conversely, an increase to pH 7, resulted in the detection of shorter oligomers by HRMS, accompanied by a reduction in water elimination (Supplementary Table 34). This outcome is consistent with the notion that enamine formation and water elimination are favoured under acidic conditions.

When adding the salts NaCl, $MgCl_2$ or $CaCl_2$ to simulate an early ocean environment resulted in an increased reaction rate, enhanced water elimination, and the formation of longer oligomers (Supplementary Table 16). However, the presence of high concentrations of NaCl led to a significant inhibition of the self-assembly process, as observed by light microscopy analysis. Conversely, the organocatalytic system exhibited persistent assembly in reactions with $MgCl_2$ and $CaCl_2$, suggesting that it functions optimally at lower salt concentrations. The simulation of an early ocean environment by the adding NaCl, $MgCl_2$ or $CaCl_2$ resulted in an increased reaction rate. Furthermore, the assemblies tolerated $MgCl_2$ and $CaCl_2$ (Supplementary Fig. 7). In comparison, fatty acids or phosphates are more susceptible to $M^{2+}$ metal ions as membrane formation is generally disrupted by them[1,43,44].

To ascertain the robustness of our reaction system over time, a wide range of conditions were monitored by time resolved HRMS (Supplementary Tables 4-7) and microscopy (Supplementary Tables 11−14). To achieve higher reaction rates over the observed time span, higher concentrations, i.e. 1 M acetaldehyde and 10 mol%

catalyst, were selected as a suitable model system. This facilitated the observation of the reaction progressing from a state of low particle count to an exponential increase over the course of 1 d (Fig. 3a, see also Supplementary Movie 2). During this process, an increase in the maximum chain length from 14 to 20, as well as a general increase in abundance of shorter chain lengths, was observed, showing the dynamic growth of the oligomers over time (Fig. 3b). It is a well-established principle that, in general, the abundance of a given substance decreases with increasing chain length. This phenomenon can be attributed to the dynamic change of selectivity by decreasing solubility and phase transition from dissolved liquids to lipids with increasing oligomeric chain length[45]. Furthermore, an increase in oligomeric chain length over C20 was not observed in timeframes of up to 7 d, which is a remarkable result as it indicates that the selectivity of the chain length of the lipid chains observed in living biological systems between C16 and C20 may have its origin in the physical properties, namely the solubility in the water phase and the ability to form stable membranes by van-der-Waals interactions of the organic chain and self-assembly. Interestingly, a heterogeneous distribution of chain lengths can facilitate the self-assembly of vesicles[46,47]. In comparison, the aldol-oligomerisation in toluene catalysed by a cation exchange resin yields significantly larger oligomers[48]. It is noteworthy that at the inception of the reaction, the presence of hydrated chains was observed. However, over time, a shift towards a greater proportion of unsaturated lipid chains becomes evident, attributable to the irreversible elimination of water during the reaction.

To illuminate the growth of the droplets over time, a dynamic light scattering (DLS) experiment was performed. For a side-by-side comparison, we used the same reaction conditions as in Fig. 3a to determine the median hydrodynamic diameter $D_h$ of particles in solution and track changes as the reaction advances (Fig. 3b-e). Initially, predominantly assemblies of approximately 100 nm in diameter were observed, which underwent growth over 6 h up to a limit of ≈800 nm (Fig. 3c). This observation is consistent with the observation in Fig. 3a, that after 1 h nearly no macromolecular structures were observed, as the formed assemblies were yet too small for microscopic observation. Concurrently, the DLS experiment detected the emergence of larger assemblies in the µm-range, whose significant increase in numbers can also be observed in the micrographs from 6 h onward (Fig. 3d,e).

In order to further understand the emergence and properties of the nm-scale particles, transmission electron microscopy (TEM) was used. Initially, TEM negative stains were obtained of dried samples using uranyl acetate for contrast (Supplementary Fig. 15). This analysis yielded a diverse array of spheres ranging from 60 nm up to 2.0 µm in size, depending on reaction time (Fig. 4a). Negative staining gives only the outline of the assemblies and provides little information about the structure itself.

Consequently, the assembly boundaries were investigated in solution by cryo-TEM (Supplementary Figs. 16−17). The structure of the particles was visualised by vitrification of the reaction mixture (Fig. 4b, c). Here, we found spherical structures ranging in size from approximately 11 nm to 1.5 µm, which is consistent with the measurements of the negative stain and the DLS experiment. The structure depicted in Fig. 4b is characterised by a distinct boundary, though the presence of a double membrane remains undetectable. Of particular interest are the dark spheres observed within the structure, which are indicative of local water accumulation. This water accumulation is a significant indication of the formation mechanism of the protocells formed (vide infra). These results suggest the formation of non-polar oil droplets in the initial phase. Furthermore, significantly smaller spheres were observed (Fig. 4c). Due to the small size (11 − 40 nm) and the resolution limit, a definitive evaluation of the boundary region remains unfeasible. However, measurements of the boundary region resulted in a theoretical thickness of 3.1 ± 0.2 nm independent of the sphere

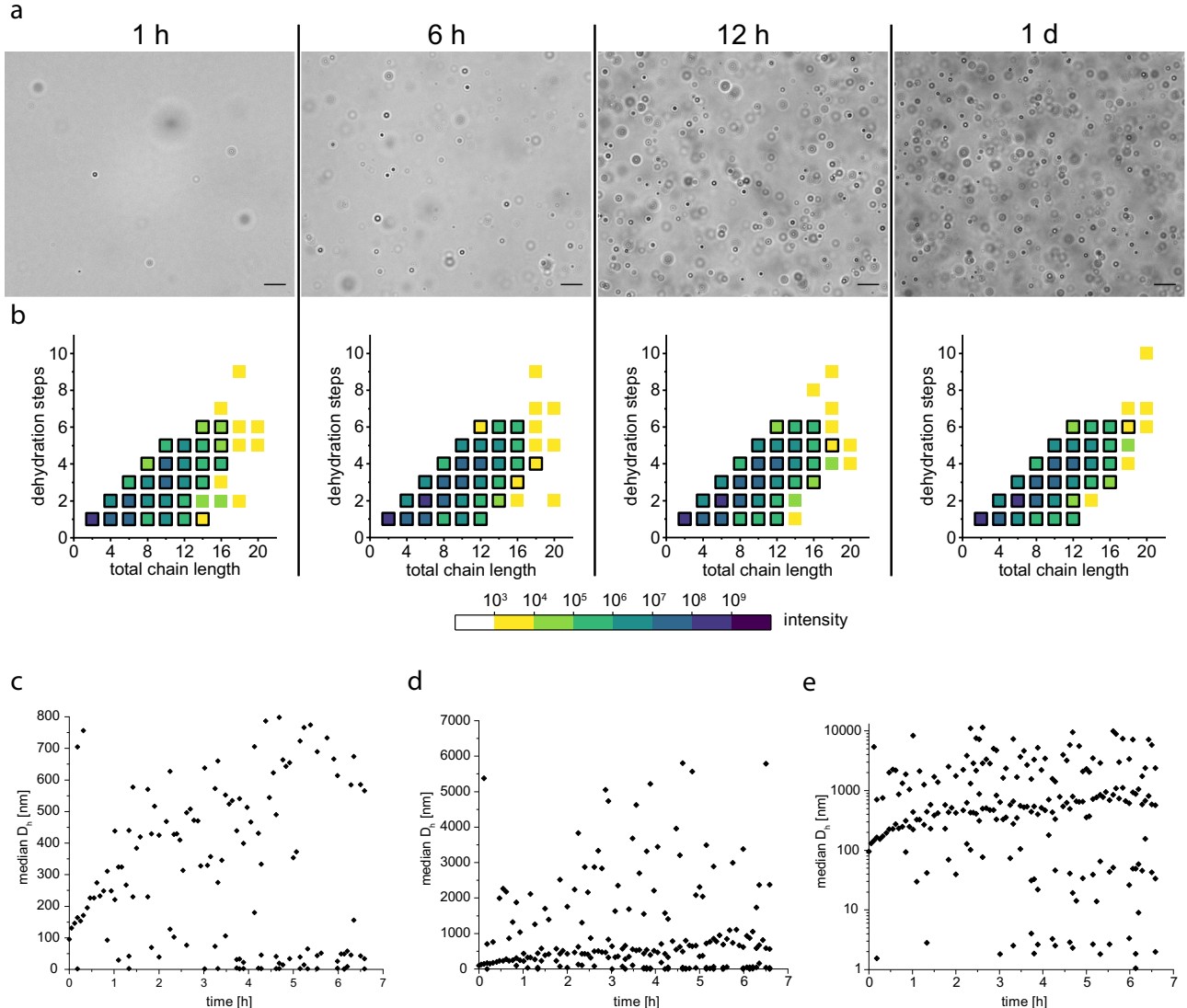

**Fig. 3 | Dynamic processes during the reaction at 1.0 M acetaldehyde with 10 mol% catalyst loading over time. a** Micrographs taken after 1 h, 6 h, 12 h and 1 d. Scale bar: 10 µm. **b** HRMS-data of catalyst chain length composition extracted for [M + H]⁺ after 1 h, 6 h, 12 h and 1 d. The reaction was repeated three times, and the intensities were averaged for every data point. For data points with a black border, the calculated mass was found in every measurement. The micrographs did not vary between samples. For a detailed figure including the variation coefficient see Supplementary Tables S8–10. **c–e** Dynamic light scattering (DLS) plots of the median hydrodynamic radius $D_h$ of particles found up to 800 nm **c**, 7 µm **d** and log-scale up to 10 µm **e**, respectively, against the time at 1.0 M acetaldehyde with 10 mol % catalyst loading. Data points smaller than 1 nm were omitted due to solvent effects.

diameter, which implicates a bilayer (Supplementary Fig. 18, Supplementary Table 36). In comparison, the bilayer formed by decanoic acid is 1.4 – 1.5 nm[49] and of phosphatidylcholines 1.5 – 3.7 nm[50]. In addition to the time-frozen cryo-TEM study, we also wanted to investigate possible dynamic processes, as permeability, accumulation and growth in the compartments are essential for the development of living cells.

Fluorescence microscopy using rhodamine B as a fluorescent probe for non-polar membranes showed that the entire sphere was uniformly illuminated (see Fig. 4d). As the dye was added immediately before the measurement, it can be concluded that the boundary was permeable to rhodamine B. In general, fluorescence is quenched less in more nonpolar environments, providing a contrast to the surrounding polar water[51–53]. This phenomenon is used, for example, to observe non-polar regions in bilayers or oil droplets[54]. As the inside of the spheres is significantly brighter, the spheres are composed of an internally uniform non-polar medium, unlike normal vesicles, which contain water. We therefore conclude that an oil droplet is the best

approximation for the found results, as deduced from the previous experiments. We propose that an interface to the surrounding water is formed by the amine of the five-membered ring of the evolved catalyst, which is protonated at the lowered pH. This head group can interact with the surrounding water, while the side chains point into the non-polar interior (Fig. 4g).

Subsequent observation of the fluorescence of the droplets over time reveals the emergence of small dark spherical areas, which grow and merge within the interior (Supplementary Movie 3, Fig. 4e). The higher fluorescence quenching suggests a more polar medium compared to pure water. This observation is plausible since during the catalytic cycle dehydration in the aldehyde/catalyst pool continuously produces water (Fig. 2c). These dark spheres were also observed in the cryo-TEM Fig. 4b (Supplementary Figs. 16–17).

The observed increase in fluorescence intensity in the non-polar phase indicates that the phase even becomes increasingly non-polar as the elimination of water and the conversion of the polyols to polyenes proceed. The formed water is immiscible with the non-polar medium

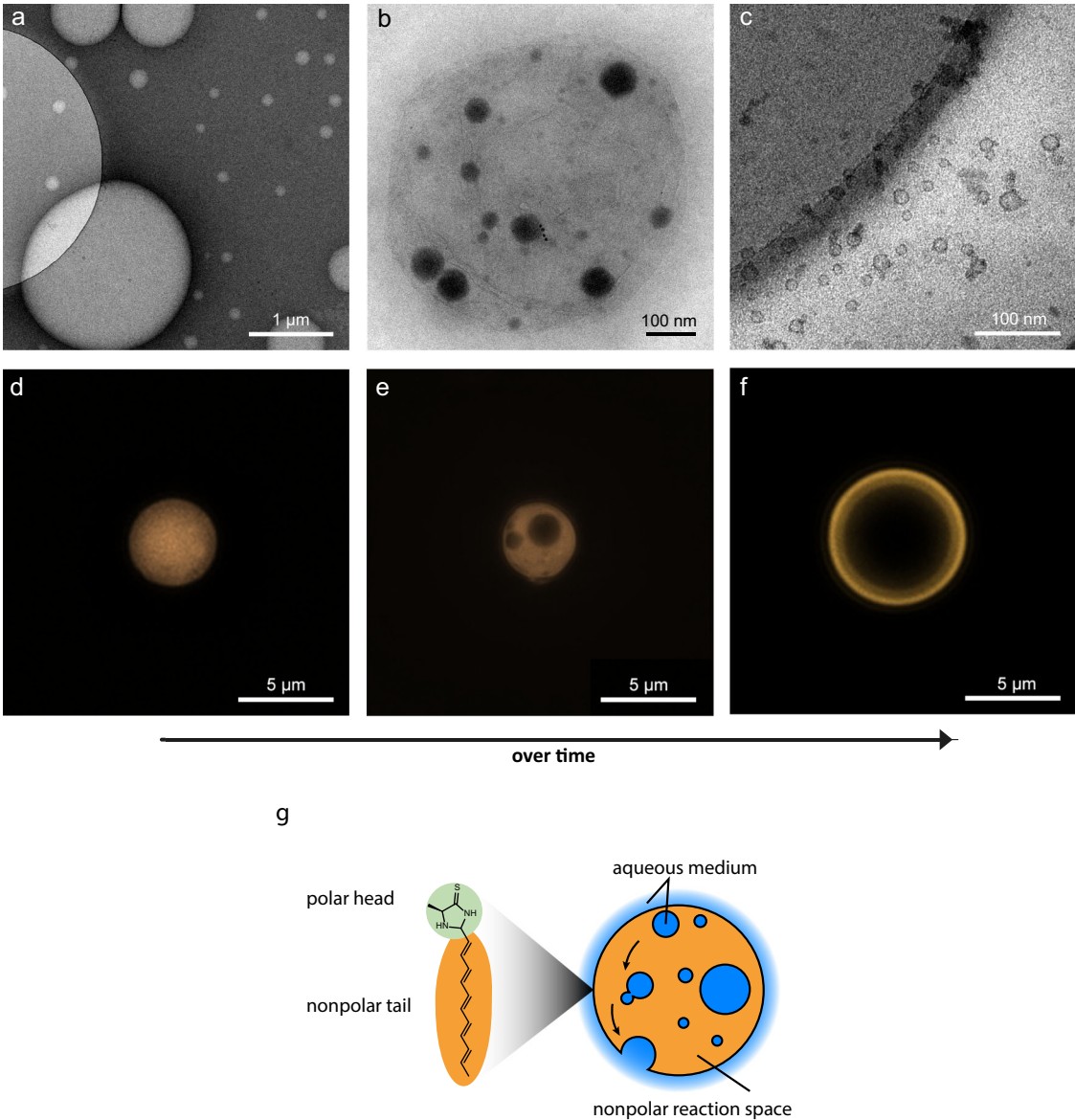

**Fig. 4 | (Cryo)-TEM micrographs and fluorescence microscopy. a** Negative stain using uranyl acetate of a reaction starting with 0.2 M acetaldehyde and 5 mol% catalyst loading after 6 h. Scale bar: 1 μm. The uniform partial circle is a hole in the holey carbon-supported grid. **b, c** Cryo-TEM image of a reaction using 0.5 M acetaldehyde with 5 mol% catalyst loading after 1 h. Scale bar: 100 nm. A water enriched oil droplet **b** and significantly smaller structures **c** are visible. **d, e,** and **f** Fluorescence microscopy of a reaction using 0.5 M acetaldehyde with 5 mol% catalyst loading developing over time. Orange: nonpolar probe rhodamine B. Scale bar: 5 μm. **d** Initial oil droplet formation. **e** droplet with water accumulation. **f** Protocell structure. **g** Proposed model for protocell formation.

and concentrates on the inside, thermodynamically driven, in a second phase as small droplets, which combine over time and are even emptied to the outside of the sphere in some cases (Supplementary Movies 4, 5). To assess the impact of local warming of the sample during fluorescence microscopy, a reaction was conducted at 40 °C (Supplementary Tables S29−30). Fluorescence microscopy revealed an increase in the overall particle size (up to 7 μm) and an accumulation of water within the sphere. Previously this phenomenon was predominantly observed during extended periods of fluorescence microscopy observations. Consequently, it can be concluded that elevated temperatures or local irradiation result in enhanced conversion, thereby facilitating water elimination. In comparison, a test reaction conducted at 0 °C did not exhibit these characteristics and light micrographs as well as HRMS measurements indicated a decelerated reaction (Supplementary Tables S32−33).

Phase separation and excretion of water play a pivotal role in the progress of a first chemical metabolism, as water is constantly removed from equilibrium. This process further facilitates the elimination of alcohols in the side chain, leading to the formation of more non-polar chains, and supports the exchange reaction of the aldehydes with the catalyst. This behaviour may have facilitated the concentration of nonpolar molecules and driven forward condensation reactions with other compounds, which are difficult in a purely aqueous system.

However, should the formed water not be shuttled outside, then all water droplets will combine over time. The resultant protocell structure exhibited encapsulation of water by a thin layer of non-polar medium (Fig. 4f). The hypothesis is thus put forward that the evolved amphiphilic catalysts adapt to form a double layer due to internal water accumulation. To test the hypothesis, the reference compound **10c + d** (condensation product of aldehyde **7** with thioamide **8**) was

dissolved at pH 4, after which vesicle-structures were observed by means of fluorescence microscopy (Supplementary Fig. 14). This finding is a so far unknown and unique process for in-situ synthesis of lipid-like molecules and simultaneous self-assembled protocell formation in a single step without external modulation of reaction conditions. In comparison, Rubio-Sanchez et al. reported on oil droplet to membrane transformation by using the oil droplet surfaces as templates[55]. This observation may provide an explanation for the transition from oil droplets via vesicle-like structures to protocells that occurred on Earth in the early stages of its evolution. This principle can be theoretically transferred to similar reaction systems. The formation of protocells marks a significant milestone in the evolution of life on Earth. Given the dye's ability to permeate the protocell, it can be deduced that other organic compounds must also be capable of doing so, leading to their accumulation within the vesicle.

In summary, a lipid oligomer formation reaction was identified as a key step in the process that leads to the self-assembly of molecules, the formation of oil droplets, and ultimately the development of protocells. This process, which commences with small molecules such as acetaldehyde, does not necessitate the use of detergents or lipid precursors. Remarkably, the lipids produced in this organocatalyzed aldol reaction and condensation cascade have lengths of up to C20, which is similar to those found in biological systems.

The organocatalytic reaction cycle was elucidated, and a mechanism involving interlocking catalytic reaction cascades was proposed. The imidazolidine-4-thione organocatalysts are pre-biotically plausible, as they are formed in high yield from small molecules (acetaldehyde, hydrogen sulphide, ammonia, and cyanides) that were abundant on early Earth. These organocatalysts exhibit a unique property: the ability to undergo molecular evolution through the catalytic modification of their own building blocks, in this case aldehyde precursors, and assimilation into their own structure.

This previously unknown process of lipid formation, self-assembly and self-modification of the initial catalyst was experimentally observed. The distinct catalytic modification of the resulting lipids leads to the catalytic activation of the entire surface of the resulting protocells. The physical properties, size, stability towards external stimuli (pH, temperature and salt content) and formation of the droplets, their transition to vesicles and protocell structures were characterised by fluorescence microscopy, DLS experiments, and cryo-TEM. The smallest structures found were about 11 nm, and the largest reached up to 7 μm. Time-resolved fluorescence experiments showed the incorporation of the dye rhodamine B and internal water production by condensation reactions, leading to the transformation of oil droplets into protocells.

This process is stable and reproducible over a wide range of reaction conditions, including varying concentrations, pH, temperature, and salts. An interesting feature of the initially formed droplets and final protocells is the accumulation of organic compounds through the lipid membrane. As a result, the concentration ratios change rapidly, as the organic molecules in these self-assemblies can be concentrated to a higher level. Once started, this feature would also allow for the reactions to progress in prebiotic environments with low concentrations of potential organic substrates. Combined with the catalytically active surface, this leads to an exponential formation of protocells, as observed in the in situ microscopic tracking of the protocell formation. These conditions may also favour condensation reactions in general, such as the oligomerisation of RNA.

Our results show that it is possible to form the first macro-molecular structures from simple molecules under abiotic reaction conditions, without the need for a harsh environment or extraterrestrial supply of materials. These protocells are a starting point for further research into the emergence of life, as they form under the simplest reaction conditions and simultaneously solve many challenging problems, such as the concentration of organic compounds, while providing a protected reaction space for the formation of life-relevant molecules.

## Methods
### Synthesis of tert-butyl (S)-(1-amino-1-thioxopropan-2-yl) carbamate 9
tert-Butyl (S)-(1-amino-1-oxopropan-2-yl)carbamate (3.76 g, 20.0 mmol, 1.00 equiv.) and LAWESSON's reagent (4.45 g, 11.0 mmol, 0.55 equiv.) were dissolved in dry THF (100 mL) and were stirred under argon for 1 h. All volatiles were removed in vacuo. The crude product was purified by flash column chromatography (n-pentane/EtOAc, 5:2 to 1:1) to obtain 9 (2.80 g, 13.6 mmol, 68%) as a white solid.

### Synthesis of (S)-2-aminopropanethioamide hydrochloride 8·HCl
9 (2.63 g, 12.9 mmol, 1.00 equiv.) was dissolved in dry DCM (40 mL) under argon. HCl in $Et_2O$ (2.0 M, 25.7 mL, 4.00 equiv.) was added and the solution was stirred for 2 h. The solution was filtered off and the precipitate was washed with dry $Et_2O$ (3 × 20 mL) under argon. The precipitate was dried under high vacuum. The product 8·HCl (1.66 g, 11.9 mmol, 92%) was obtained as a white solid and stored under argon.

### Synthesis of (5S)-2,5-dimethylimidazolidine-4-thione 1
8·HCl (148 mg, 1.05 mmol, 1.00 equiv.) was dissolved in an aqueous $NaHCO_3/Na_2CO_3$ buffer (0.2 M, 5 mL, pH 9.2) at 0 °C. Acetaldehyde (70.0 μL, 1.26 mmol, 1.20 equiv.) was added and stirred for 90 min. Then, additional acetaldehyde (205 μL, 3.68 mmol, 3.50 equiv.) was added and the reaction was allowed to warm to room temperature. After 2 h, all volatiles were removed in vacuo. The crude product was purified by flash column chromatography (EtOAc, $R_f$ = 0.20) to obtain 1 (87.4 mg, 762 μmol, 64%, $dr$ = 1.05) as a white solid.

### Synthesis of ethyl (E)-3-hydroxyhex-4-enoate 11
A solution of diisopropylamine (7.87 mL, 56.0 mmol, 1.40 equiv.) in THF (26 mL) was cooled to 0 °C. n-BuLi (2.5 M, 19.2 mL, 48.0 mmol, 1.20 equiv.) was added and stirred for 10 min. The prepared LDA solution was cooled to –78 °C. Dry ethyl acetate (4.69 mL, 48.0 mmol, 1.20 equiv.) was added and the solution stirred for 40 min. Then, (2E)-but-2-enal (3.31 mL, 40.0 mmol, 1.00 equiv.), dissolved in dry THF (5 mL), was added dropwise at –78 °C. After 1 h, aq. conc. $NH_4Cl$ (20 mL) was added slowly and the reaction was allowed to warm to r.t. over 1 h. The reaction mixture was extracted with EtOAc (3 × 150 mL) and dried over $MgSO_4$. The crude product was purified by flash column chromatography (c-Hex/EtOAc 7:1, $R_f$ = 0.24) to obtain 11 in quantitative yield (6.33 g, 40.0 mmol, 100%) as a colourless oil.

### Synthesis of ethyl (E)-3-((tert-butyldimethylsilyl)oxy)hex-4-enoate 12
Imidazole (9.26 g, 136 mmol, 3.4 equiv.) was dissolved in dry DCM (70 mL) and TBDMSCl (7.36 g, 48.8 mmol, 1.22 equiv.), dissolved in dry DCM (20 mL), was added. Ethyl (E)-3-hydroxyhex-4-enoate 11 (6.33 g, 40.0 mmol, 1.00 equiv.), dissolved in dry DCM (20 mL), was added to the white suspension and stirred for 1 h. The reaction mixture was quenched with sat. aq. $NaHCO_3$ (100 mL) and extracted with DCM (3 × 150 mL). The solution was dried over $Na_2SO_4$ and concentrated. The crude product was purified by flash column chromatography (c-Hex/EtOAc 7:1, $R_f$ = 0.76) to obtain the product 12 in quantitative yield (10.9 g, 40.0 mmol, 100%) as a colourless oil.

### Synthesis of (E)-3-((tert-butyldimethylsilyl)oxy)hex-4-enal 13
12 (822 mg, 3.02 mmol, 1.00 equiv.) was dissolved in dry THF (15 mL) and cooled to –90 °C. DIBALH in THF (1 M, 5.25 mL, 5.25 mmol, 1.74 equiv.) was added dropwise and the solution was kept at –90 °C. Sat. aq. $NH_4Cl$ (1.5 mL) was added after 2.5 h while keeping the temperature below –75 °C. After allowing to warm to r.t., a sat. aq. Na,K-tartrate solution (50 mL) was added. The reaction mixture was extracted with

EtOAc (5 × 40 mL), dried over $Na_2SO_4$ and concentrated. The crude mixture was purified by flash column chromatography (*n*-pentane/ EtOAc 98:2, $R_f$ = 0.3) to obtain **13** (444 mg, 1.96 mmol, 65%) as a colourless oil.

### Synthesis of (5*S*)-2-((*E*)-2-((tert-butyldimethylsilyl)oxy)pent-3-en-1-yl)-5-methylimidazolidine-4-thione 14

**8-HCl** (1.13 g, 8.00 mmol, 1.00 equiv.) was dissolved in a $NaHCO_3$/ $Na_2CO_3$ buffer solution (0.2 M, pH 9, 20 mL) and added to a solution of **13** (1.83 g, 8.00 mmol, 1.00 equiv.) in MeCN (20 mL). After 2 h, all volatiles were removed in vacuo and the crude product was extracted with DCM (5 × 50 mL). The organic phase was dried over $Na_2SO_4$, concentrated and purified by flash column chromatography (*n*-pentane/EtOAc 3:1). Three fractions were collected containing the product diastereomers, whereby the third fraction contained two isomers. **14a** (22.8 mg, 0.07 mmol, 0.9%), **14b** (28.3 mg, 0.09 mmol, 1.1%) and **14c** + **d** (70.1 mg, 0.22 mmol, 2.8%).

### Synthesis of (2*S*,5*S*)-2-((*R*,*E*)-2-hydroxypent-3-en-1-yl)-5-methyl-imidazolidine-4-thione 10a

**14a** (21.5 mg, 68 µmol, 1.00 equiv.) was dissolved in dry THF (0.1 M, 0.68 mL) and (*n*Bu)$_4$NF in THF (1 M, 78 µL, 78 µmol, 1.15 equiv.) was added at 0 °C. After 2 h, TMSOMe (47 µL, 341 µmol, 5 equiv.) was added and the solution stirred for 30 min. All volatiles were removed in vacuo and the crude product was purified by flash column chromatography (*n*-pentane:EtOAc 1:3, $R_f$ = 0.2). **10a** (20.8 mg, 104 µmol, 65%) was obtained as a yellow solid.

### Synthesis of (2*R*,5*S*)-2-((*S*,*E*)-2-hydroxypent-3-en-1-yl)-5-methyl-imidazolidine-4-thione 10b

**14b** (11.4 mg, 36 µmol, 1.00 equiv.) was dissolved in dry THF (0.1 M, 0.36 mL) and (*n*Bu)$_4$NF in THF (1 M, 42 µL, 42 µmol, 1.15 equiv.) was added at 0 °C. After 2 h, TMSOMe (25 µL, 181 µmol, 5 equiv.) was added and the solution stirred for 30 min. All volatiles were removed in vacuo and the crude product was purified by flash column chromatography (*n*-pentane:EtOAc 1:3, $R_f$ = 0.2). **10b** (3.5 mg, 17.5 µmol, 48%) was obtained as a yellow solid.

### Synthesis of (2*R*,5*S*)-2-((*R*,*E*)-2-hydroxypent-3-en-1-yl)-5-methy-limidazolidine-4-thione 10c and (2*S*,5*S*)-2-((*S*,*E*)-2-hydroxypent-3-en-1-yl)-5-methylimidazolidine-4-thione 10 d

**14c** + **d** (50.4 mg, 160 µmol, 1.00 equiv.) were dissolved in dry THF (0.1 M, 1.60 mL) and (*n*Bu)$_4$NF in THF (1 M, 185 µL, 185 µmol, 1.15 equiv.) was added at 0 °C. After 2 h, TMSOMe (110 µL, 801 µmol, 5 equiv.) was added and the solution was stirred for 30 min. All volatiles were removed in vacuo and the crude product was purified by flash column chromatography (*n*-pentane:EtOAc 1:5, $R_f$ = 0.45). **10c** + **d** (20.8 mg, 104 µmol, 65%) were obtained as yellow solids.

### pH Titration of (2*RS*,5*S*)−2,5-dimethylimidazolidine-4-thione 1

**1** (6.5 mg, 50.0 µmol, 1.00 equiv.) was dissolved in aq. NaOH (0.1 M, 1.00 mL, 50.0 µmol, 1.00 equiv.). The initial pH was measured, and the solution was titrated in 2.5 µL or 5 µL steps with aq. HCl (1.0 M). The pKa values were determined using the second derivative of the titration curve. For the thiolactam, we found a $pKa_1$ of 3.8, and for the secondary amine, a $pKa_2$ of -11.1.

### GP1: Organocatalyzed aldol oligomerization of acetaldehyde catalysed by 1

The catalyst **1** (0.5 mol% – 20 mol%) was added and dissolved in $H_2O$, which was adjusted to pH 4 with acetic acid. When the solution turned clear, acetaldehyde (0.2 M–2.0 M) was added, the vial was tightly closed, and the solution was stirred at room temperature for up to 7 d.

### HRMS Experiments – sample preparation and analysis

For a normalized initial catalyst (**1**) concentration of 0.1 µmol/mL during the measurement, the sample volume taken was adjusted. The sample was diluted to 100 µL with ultrapure $H_2O$ and 900 µL MeCN was added. The solution was filtered by syringe filtration using a 0.45 µm cellulose acetate *luer lock* syringe filter (red, Ø = 13 mm). The prepared solution with 0.1 µmol/mL initial catalyst (**1**) concentration was loaded into a 5 µl sample loop and injected for HRMS (ESI) measurement.

### Dynamic light scattering (DLS)

The reaction was prepared according to **GP1**, then the reaction mixture was filtered by syringe filtration (0.45 µm cellulose acetate luer lock syringe filter, red, Ø = 13 mm). The solution was transferred to a cuvette and the measurement was started. For data acquisition, a VISCOTEK 802 DLS setup was used with the cell temperature set to 20 °C. The laser intensity was automatically adjusted to 300,000 counts prior to every data collection. Every 5 min, a new data acquisition was started. The data was analysed using the software *OmniSIZE* version 3.0.0.296 from VISCOTEK. Datapoints smaller than 1 µm were omitted due to solvent effects.

### (Fluorescence) microscopy

A ZEISS Axioscope 5 was used in combination with a monochromatic camera Axiocam 705 mono, mounted with a 0.65× adapter. Unless stated otherwise, a 100× objective EC Plan-Neofluar 100×/1.30 Oil was used. Immersol™ 518 F was used for oil immersion microscopy to increase resolving power. For sample preparation, 80 µL of the reaction mixture was added to the cavity of a microscope slide (76 × 26 × 1.35 mm with 2 cavities Ø = 15–18 mm, depth 0.6–0.8 mm, from MARIENFIELD) and covered with a cover glass (thickness no. 1½, high-performance, 18 mm × 18 mm, 0.170 ± 0.005 mm, from ZEISS).

For fluorescence microscopy, a HXP 120 V fluorescence light source was used in combination with the filter set 43 HE from ZEISS. Excitation: BP 550/25 (HE), Beamsplitter: FT 570 (HE) and Emission: BP 605/70 (HE). For sample preparation, 90 µL of the reaction mixture was mixed with 10 µL of the fluorescent dye rhodamine B (1 µM), then 80 µL was taken and added to the cavity of a microscope slide and covered with a cover glass.

## Data availability

All data generated or analysed during this study are included in this published article (and its Supplementary Information Files), and are available from the corresponding author upon request. The X-ray crystallographic coordinates for structure 10c+d reported in this study has been deposited at the Cambridge Crystallographic Data Centre (CCDC), under deposition number CCDC-2477377. These data can be obtained free of charge from The Cambridge Crystallographic Data Centre via www.ccdc.cam.ac.uk/data_request/cif.

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

## Acknowledgements
We thank the Max - Planck Society (Max - Planck - Fellow Research Group 'Origins of Life', OT), Germany's Excellence Strategy, ORIGINS, EXC – 2094 – 390783311 (OT), DFG/German Research Foundation, Project – ID 521256690 – TRR 392, Molecular Evolution (OT), and the Volkswagen Stiftung, Initiating Molecular Life (OT) for funding.

## Author contributions
M.S.R.E. and O.T. conceived and designed the experiments. M.S.R.E. and K.H.N. performed the experiments. M.S.R.E. and O.B. performed microscopy and (cryo-) TEM experiments. M.S.R.E., O.B., R.B., and O.T. analysed the data. M.S.R.E., O.B., R.B., and O.T. wrote the paper. All authors discussed the results and commented on the manuscript.

## Funding

## Competing interests
The authors declare no competing interests.
