## [Transparent Peer Review file · Nature Communications]

Organocatalyzed Bottom-up Formation of Protocells

Corresponding Author: Professor Oliver Trapp

Version 0:

Reviewer comments:

Reviewer #2

(Remarks to the Author)

The authors have addressed many of the reviewers' comments. While a more complete characterisation of the synthesised lipids, catalysis kinetics, and generated membranes would have been appreciated, this would have required months of work and infrastructure/expertise that might not be directly available to the authors. I believe the manuscript is suitable for publication in its current form (after a few typos are corrected here and there).

Reviewer #3

(Remarks to the Author)

The authors have thoroughly revised the manuscript and successfully addressed all of my previous comments and suggestions. The revisions are comprehensive, and all changes have been implemented. The updated version shows clear improvement in clarity, structure, and scientific rigor. The responses are detailed and clear, I appreciate the authors' thoughtful engagement with the feedback. Overall, the paper now meets the standards for publication, and I recommend it for acceptance.

List of Changes Manuscript Nature Communications NCOMMS-25-80741-T

On behalf of all the authors, I would like to thank the competent reviewers for providing us with feedback on our manuscript. We greatly appreciate all the helpful suggestions and valuable comments provided by the reviewers to improve the quality of the manuscript.

Reviewer #2 (Remarks to the Author):

The authors have addressed many of the reviewers' comments. While a more complete characterisation of the synthesised lipids, catalysis kinetics, and generated membranes would have been appreciated, this would have required months of work and infrastructure/expertise that might not be directly available to the authors. I believe the manuscript is suitable for publication in its current form (after a few typos are corrected here and there).

Thank you very much for the very kind comments!

Reviewer #3 (Remarks to the Author):

The authors have thoroughly revised the manuscript and successfully addressed all of my previous comments and suggestions. The revisions are comprehensive, and all changes have been implemented. The updated version shows clear improvement in clarity, structure, and scientific rigor. The responses are detailed and clear, I appreciate the authors' thoughtful engagement with the feedback. Overall, the paper now meets the standards for publication, and I recommend it for acceptance.

Thank you very much for the very kind comments!